# Bending Stiffness, Load-Bearing Capacity and Flexural Rigidity of Slender Hybrid Wood-Based Beams

**Barbara Šubic [1],\*, Gorazd Fajdiga [2] and Jože Lopatič [3]**

1   M SORA d.d., Trg Svobode 2, 4226 Žiri, Slovenia
2   Biotechnical Faculty, University of Ljubljana, Jamnikarjeva 101, 1000 Ljubljana, Slovenia;
    Gorazd.Fajdiga@bf.uni-lj.si
3   Faculty of Civil and Geodetic Engineering, University of Ljubljana, Jamova 2, 1000 Ljubljana, Slovenia;
    Joze.Lopatic@fgg.uni-lj.si
\*   Correspondence: Barbara.Subic@m-sora.si; Tel.: +386-31-541-681

**Abstract:** Modern architecture suggests the use of opened spaces with large transparent envelope surfaces. Therefore, windows of long widths and large heights are needed. In order to withstand the wind loads, such wooden windows can be reinforced with stiffer materials, such as aluminium (Al), glass-fibre reinforced polymer (GFRP), and carbon-fibre reinforced polymer (CFRP). The bending stiffness, load-bearing capacity, and flexural rigidity of hybrid beams, reinforced with aluminium, were compared through experimental analysis, using a four-point bending tests method, with those of reference wooden beams. The largest increases in bending stiffness (29%–39%), load-bearing capacity (33%–45%), and flexural rigidity (43%–50%) were observed in the case of the hybrid beams, with the highest percentage of reinforcements (12.9%—six reinforcements in their tensile and six reinforcements in their compressive zone). The results of the experiments confirmed the high potential of using hybrid beams to produce large wooden windows, for different wind zones, worldwide.

**Keywords:** wood based composites; hybrid beams; bending stiffness; flexural rigidity; aluminium reinforcements; wooden windows

## 1. Introduction

The share of wood-based windows (i.e., wooden and aluminium-wooden profiled windows) is the largest (66%) among certified windows for low-energy houses, due to their high energy efficiency and their beneficial life cycle assessment characteristics [1]. Polyvinyl chloride (PVC) and aluminium windows have 19% and 13% shares, respectively. Contemporary architectural design of buildings integrate wooden products in the building envelope as well as in the interior. One of the reasons for this is that it was proven that the use of natural wood in the buildings has positive psychological, emotional, and health impacts on the people living in such an environment [2]. Windows have an important role (solar gains through glazing), effecting the energy efficiency of the building. Therefore, there has been a gradual increase in the proportion of the transparent part of the building envelope, over the last decade. Windows are more and more frequently placed from the bottom to the top of individual storeys, with heights exceeding 3.0 m, or even over several storeys, with heights exceeding 5.0 m (an example of such building is presented in Figure 1).

Two most important window elements are window frame and window glazing. From a thermal point of view, the window frame is a weak point of the low energy windows, since its thermal transmittance varies from 0.8–1.6 W/m$^2$K, whereas thermal transmittance of the glazing is almost half this value and varies from 0.5–0.8 W/m$^2$K [1]. To improve thermal transmittance of the window

we need to either improve thermal transmittance of the frame or increase the surface of the glazing, compared to the whole window area. The architectural trends prefer the open view through windows, with as little visible window frame as possible. Therefore, for this study small cross sectioned (i.e., slender) profiles were analysed, to explore the limits of improving their bending stiffness and strength, when being reinforced with stiffer materials.

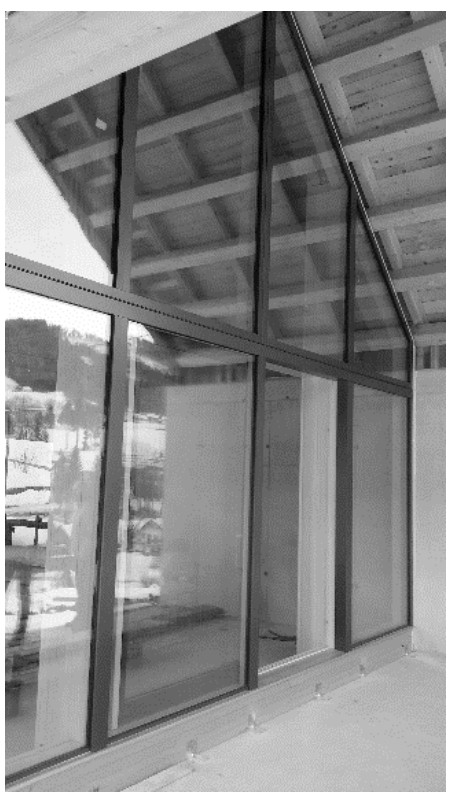

**Figure 1.** An example of window elements with a height of 5.8 m.

In Europe, windows are mostly manufactured from softwood species—90% (e.g., Norway spruce—*Picea abies* (L.) H. Karst., Pine—*Pinus sylvestris* L. and Siberian larch—*Larix sibirica* Ledeb.), whereas hardwoods are seldom used—10% (e.g., oak—*Quercus* spp. L., Red grandis—*Eucalyptus grandis* W. Hill ex Maiden, walnut—*Juglans regia* L., beech—*Fagus sylvatica* L.) [3]. The main reason for this is that softwoods (especially Norway spruce and Pine, which in Europe are a domestic wood species), are easy to process and have up to 38% better thermal properties than hardwoods [4]. This is an important parameter when trying to comply with today's high thermal efficiency demands for modern buildings.

Over the last twenty years, many studies have been performed with regard to the flexural rigidity and load-bearing capacity of hybrid beams; not only experimentally [5–8], but also analytically [9–14] and numerically [15–19]. In most of the mentioned studies much larger cross sections than those used for windows were investigated (e.g., up to 500 mm in height) [12,15]. Only a few researchers have studied the reinforcing effect on specimens with cross sections smaller than 100 mm × 100 mm [14,19,20], where limited space for reinforcements is available. In most cases the reinforcements are visible at least on one side of the tested specimens and mostly oriented horizontally [5,6,12]. Windows are influencing the overall architectural appearance of the building from the outside, as well as the interior design, and therefore no visible reinforcements are preferred on either side of the window.

Different reinforcing materials have been used in many studies to improve the bending stiffness and load-bearing capacity of wooden beams. Steel profiles [5,9,12], glass-fibre reinforced polymer profiles (GFRP (glass-fibre reinforced polymer) profiles) [6,14,18], and carbon-fibre reinforced polymer profiles (CFRP (carbon-fibre reinforced polymer) profiles) [7,11,15,19] are the most common, but the

effect of many others has also been investigated; e.g., basalt fibres [13], hemp, flax, basalt, and bamboo fibre reinforcements [21], as well as concrete in combination with wood [22].

The aim of this study is to explore possibilities of using domestic wood species for technologically challenging big-sized window elements, with high added value. For this reason, the Norway spruce, as the basic wood species, was used in this study.

During the first stage of the study, the mechanical characteristics of the used materials were defined. Slender hybrid beams were manufactured using aluminium reinforcements and four-point bending tests were performed in order to compare the bending stiffness and load-bearing capacities of beams having different percentages, orientations, and positions of the reinforcing material.

All decisions about the type of adhesive, the number and position of the reinforcements and the choice of material were made taking into account not only the required mechanical characteristics but also future production feasibility and cost-effectiveness.

## 2. Materials and Methods

Norway spruce was used as the basic material for all the beam specimens which were prepared for this study. Beams were manufactured from lamellas with thicknesses from 8 to 28 mm, with a measured averaged moisture content of 12% and an average density of 460 kg/m$^3$. The mechanical characteristics of the spruce, that correspond to structural grade C30 [23], used in window production, compared to those of the reinforcing materials, are presented in Table 1. The shear strength was determined experimentally, according to standard EN 205 [24].

**Table 1.** Material characteristics.

|  | Norway Spruce | Aluminium | GFRP | CFRP |
|---|---|---|---|---|
| Density $\varrho$ (kg/m$^3$) | 460 | 2700 | 2000 | 1420 |
| Poisson's ratio $\nu$ | 0.37 | 0.30 | 0.27 | 0.3 |
| Modulus of elasticity E$_0$ (MPa) | 12,000 | 70,000 | 38,000 | 120,000 |
| Shear modulus G (MPa) | 750 | 25,000 | 5000 | 4000 |
| Tensile strength f$_t$ (MPa) | 19 | 195 | / | / |
| Compressive strength f$_c$ (MPa) | 24 | / | / | / |
| Shear strength f$_v$ (MPa) | 8.4 * | / | / | / |
| Thermal conductivity $\lambda$ (W/mK) | 0.11 | 210 | 0.3 | 5 |
| Price factors compared to wood (per m$^3$) | 1 | 28 | 71 | 180 |

* Experimentally defined value.

Three different types of reinforcing material were planned to be used later on within the wider scope of this study; aluminium, GFRP, and CFRP profiles. The longitudinal modulus of elasticity of the CFRP profiles, aluminium, and GFRP profiles compared to that of Norway spruce are higher by a factor of 10.0, 5.8, and 3.2, respectively (Table 1). Thus, CFRP profiles should provide the best improvements in bending stiffness and load-bearing capacity [10,14,25]. Despite this, we decided to perform the first phase of the tests with aluminium reinforcements only, in order to define their positioning and orientation effect on the observed parameters and to analyse the possibilities of manufacturing window profiles with non-visible reinforcements. The advantages of aluminium are the uniformity of its mechanical properties, its high tensile capacity, and the easy processing, accessibility, and low costs of the material itself. On the other hand, its high thermal conductivity decreases the low energy efficiency of the windows, so it would be expected that in a further step of the research it would be replaced with GFRP or CFRP profiles.

The aluminium profiles used for the analysis were made of aluminium alloy 6060-T66 and technical data are collected from technical sheets provided by the manufacturer [26]. For GFRP and CFRP profiles the data were also obtained from technical data sheets from the manufacturers [27].

For this study pre-analysis of the adhesive selection for connection between aluminium and wood was performed. Tensile shear strength of three two-component epoxy adhesives (2K-EP) and three

two-component polyurethane adhesives (2K-PU) were defined according to EN 1465:2009 [28], with tensile lap-joint tests on aluminium specimens. For each adhesive, 10 specimens were tested and the shear strength was measured. The mean shear strength of the Novasil P-SP adhesive was the lowest and of COSMO PU was the highest (Table 2).

**Table 2.** Adhesives characteristics and experimental results of their shear strength.

| Brand Name | Type | Mean Shear Strength (MPa) | Standard Deviation of Shear Strength (MPa) | Technical Sheet Shear Strength (MPa) |
|---|---|---|---|---|
| Körapox 565/GB | 2K-EP | 9.9 | 0.96 | 24 |
| Permabond ET515 | 2K-EP | 4.4 | 0.53 | 10 |
| COSMO EP-200.110 | 2K-EP | 6.1 | 0.75 | 18 |
| Körapur 790/30 | 2K-PU | 8.9 | 1.13 | 18 |
| Novasil P-SP 6944 | 2K-PU | 4.0 | 0.96 | / |
| COSMO PU-200.280 | 2K-PU | 11.2 | 0.75 | 18 |

Due to its largest shear strength, it was decided to use the two-component polyurethane adhesive—COSMO PU 200.280—for further experimental analysis. All of the mean shear strengths defined through experimental analysis had a value that was lower compared to the declared ones on the manufacturers' technical sheets. Nevertheless, the shear strength of 11.2 MPa (of the selected 2K-PU adhesive), was higher than the shear strength of the wood itself (8.4 MPa), which should present sufficient strength to avoid the cohesive failure of the joint.

In the second step the four different window beam sections (scantlings), designated types A, B, C, and D, were selected for the experimental study (Figure 2). Norway spruce was used as the basic material, and aluminium as the reinforcing material.

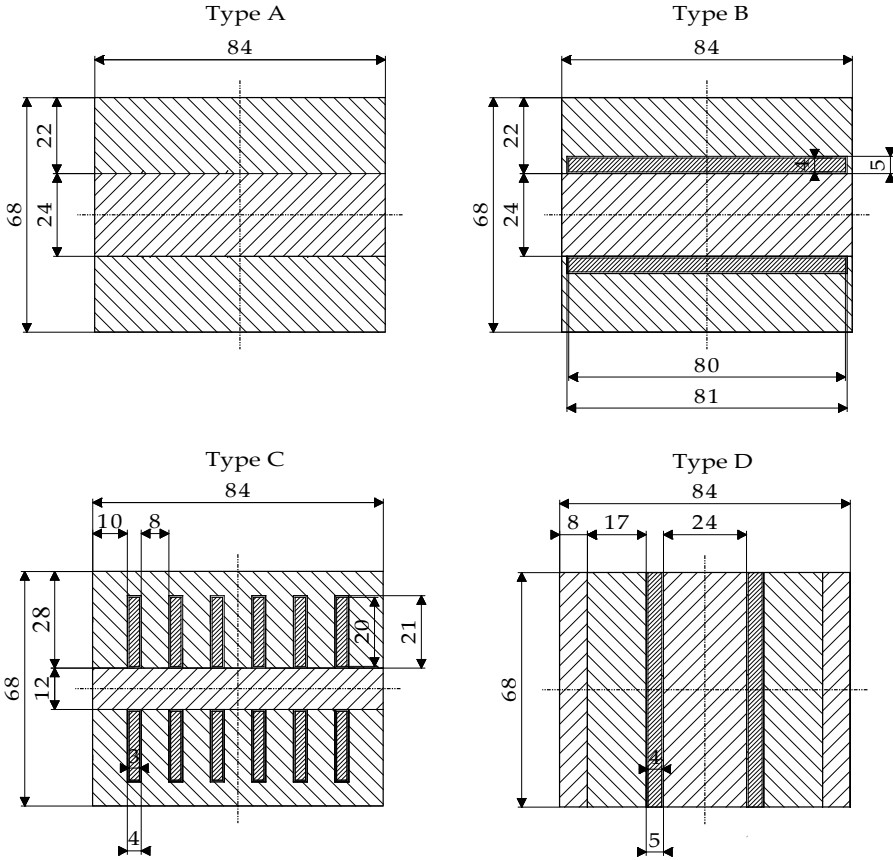

**Figure 2.** Cross sections of the type A, B, C, and D specimens (measurements in mm).

The position and orientation of the reinforcements in the specimens, B, C, and D, were defined, taking into account number of different aspects:

- To improve bending stiffness, load-bearing capacity and flexural-rigidity the reinforcements need to be positioned near tensile and compressive faces, as far away from the neutral axis as possible [9,15,25,29].
- The moment of inertia of the cross section should be maximized by proper orientation of the reinforcements and the provision of a sufficient number of them.
- An aesthetic requirement—no reinforcement should be visible when the window is completed.
- The dimensions of the reinforcements should correspond to those of standard products available on the market, in order to decrease production costs.
- The production feasibility aspect—the basic thickness of wooden lamellas, to produce window scantlings, is 24 mm.
- Cost-effectiveness of the hybrid beams.

The outside measurements of the specimens were prepared with the following dimensions: width (w) × height (h) × length ($L_e$) = 84 mm × 68 mm × 1950 mm. Basic wooden beams for specimens were manufactured from lamellas with thicknesses from 8 to 28 mm. The type A specimens were unreinforced (i.e., reference specimens). The type B specimens had two horizontal reinforcements (with width × height dimensions of 80 mm × 4 mm) placed in the pre-milled grooves of the contact regions between the two neighbouring wooden lamellas. The type C specimens had twelve vertical reinforcements (with width × height dimensions of 3 mm × 20 mm), which were located in each of the two outside wooden lamellas (six in each lamella). Type D specimens were prepared with two vertical reinforcements with dimensions of 68 × 4 mm. All the pre-milled grooves were in width and height 1 mm bigger than reinforcements, leaving 0.5 mm space for the adhesive around the reinforcements. The adhesive was applied to the grooves and the reinforcements were then manually inserted into them. Specimens C had the highest number of reinforcements with its highest area (12.6%), while specimens D had smallest number of reinforcements with the smallest area (9.5%) (Table 3).

**Table 3.** Characteristics of the tested specimens.

| Specimen | Number of Specimens (-) | Series | Dimensions of Reinforcement b × h (mm) h | Number of Reinforcements (-) | $A_{real}$ Percentage of Reinforcement (%) |
|---|---|---|---|---|---|
| A1–A5 | 5 | S1 | - | - | - |
| * B1–B5 | 4 | S1 | 80 × 4 | 2 | 11.2 |
| C1–C5 | 5 | S1 | 3 × 20 | 12 | 12.6 |
| A6–A10 | 5 | S2 | - | - | - |
| B6–B10 | 5 | S2 | 80 × 4 | 2 | 11.2 |
| C6–C10 | 5 | S2 | 3 × 20 | 12 | 12.6 |
| D1–D5 | 5 | S2 | 4 × 68 | 2 | 9.5 |

* B4 was excluded from the analysis due to its different dimension of the cross section (84 mm × 65 mm).

A decision about the preparation of type D specimens was made after the first tests of Series S1 on specimens A, B, and C had been completed, in order to be able to make additional comparisons of the effect of different reinforcement orientations on the bending stiffness, load-bearing capacity, flexural rigidity, and type of failure mechanism. The type D specimens did not fulfil the aesthetic requirement for non-visibility of the reinforcements.

Two test fields, for four-point bending tests, were prepared for two series of hybrid specimens; Series S1 (Figure 3) and Series S2 (Figure 4). The difference between them was in different spans L, lengths of the middle areas l, and the distance of the applied load from the support a. The simply supported beams of Series S1 were 1950 mm long and had a span of L = 1750 mm. Loads of P/2, which were 750 mm apart, were applied at a distance a = 500 mm from the beam supports (Figure 3). The simply supported beams of Series S2 were 1950 mm long and had a span of L = 1850 mm. In this

case loads of P/2, which were 700 mm apart, were applied at a distance a = 625 mm from the beam supports (Figure 4). The difference between both series was that the distance between the loads l decreased and the span L increased, in the case of Series S2. With this change, the bending moment with specimens of Series S2 increased, at the same loading force, compared to specimens of Series S1. This step was made in order to decrease the probability that the failure would be caused by shear strength, prior the bending strength would be fully exploited.

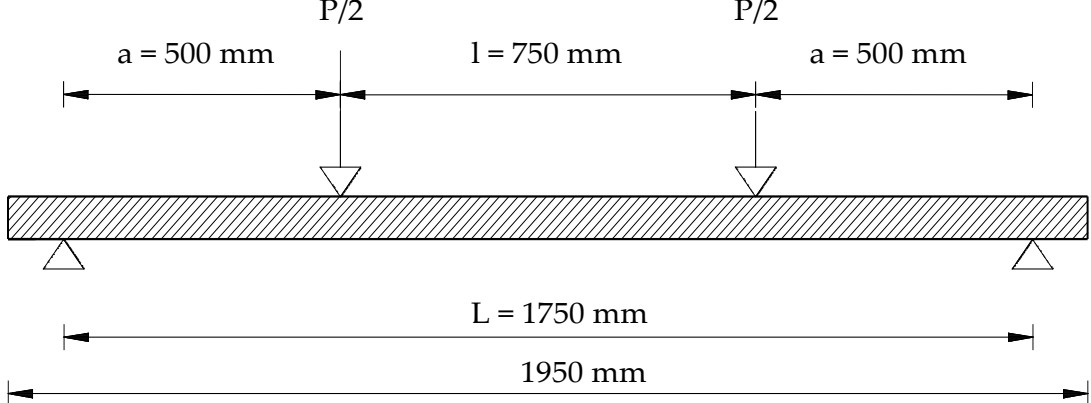

**Figure 3.** Schematic diagram of the four-point bending test field configuration for the specimens of Series S1.

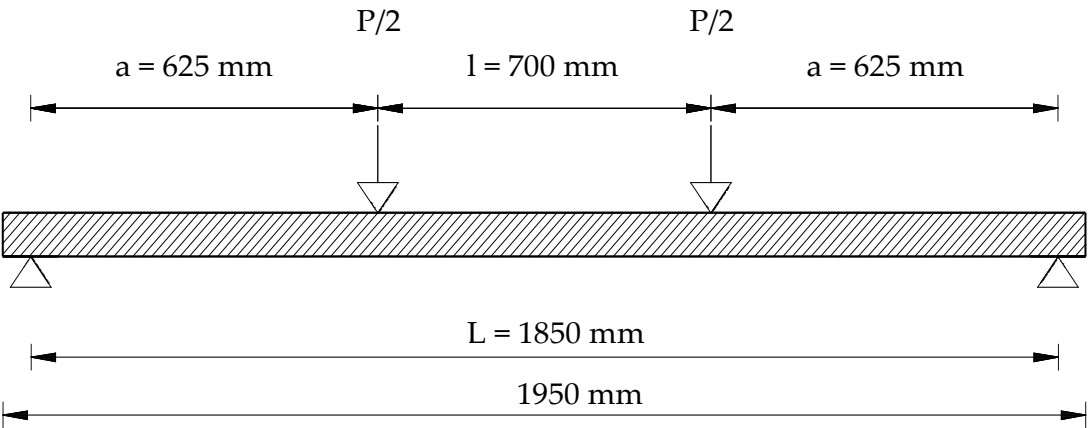

**Figure 4.** Schematic diagram of the four-point bending test field configuration for the specimens of Series S2.

The four-point static bending tests, up to failure, were performed using a Roel Amsler HA 100 universal servo-hydraulic testing machine (Zwick GmbH & Co. KG, Ulm, Germany) (Figure 5).

The flexural loading tests were performed in displacement (i.e., stroke) control mode, with an actuator movement rate of 0.1 mm/s, up to failure. In the region between the two applied loads the bending moment was constant, the shear force was zero, and the maximum relative deflection was caused by the bending moments only. On each specimen the deflections were measured at two points using linear voltage displacement transducers (LVDTs) marked as IND in Figure 6.

The absolute deflection $w_1$ was calculated as the average value of the deflections measured at the midspan by the IND 1 and IND 2. All the physical properties (i.e., the displacements, the strains, and the load P) were measured and recorded by a Dewesoft DEWE 2500 data acquisition system (DEWESoft d.o.o., Trbovlje, Slovenija).

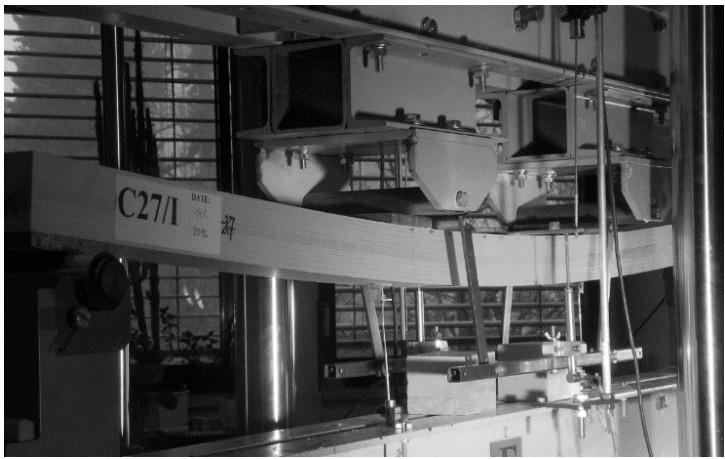

**Figure 5.** The testing equipment for the performance of the four-point bending tests.

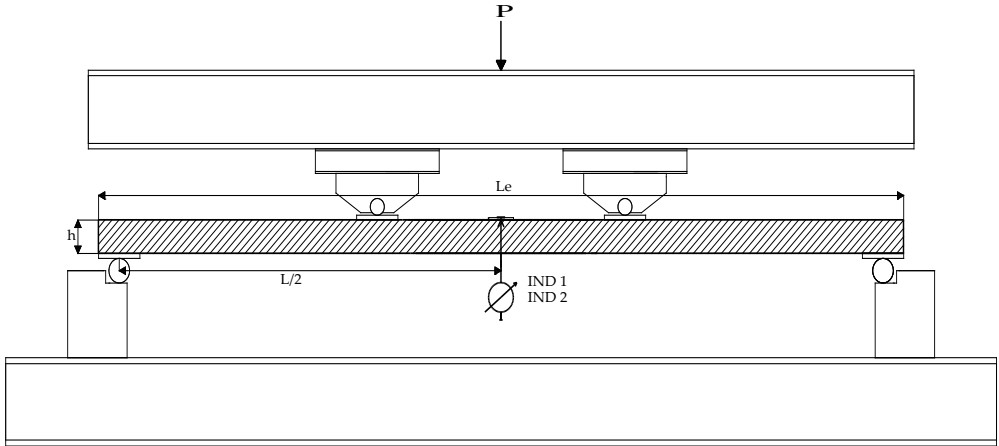

**Figure 6.** Basic configuration of the test instrumentation.

Effective flexural rigidity (EI)$_{eff}$ was calculated on the basis of the average absolute midspan deflection $w_1$, using an expression of technical mechanics (Equation (1)) that corresponds to the four-point bending test load arrangement. The equation neglects a small part of deflections caused by shear stresses.

$$(EI)_{eff} = 0.33\ P_u \cdot a \cdot (3L^2 - 4a^2)/48 \cdot w_1 \tag{1}$$

The ultimate load (P$_u$) is the load where the failure occurred, a is the distance between the support and the load, L is the span length and $w_1$ is the absolute midspan deflection.

## 3. Results

The bending stiffness was calculated for each specimen type at three different loading stages: at a load corresponding to 33% of the ultimate load (0.33 P$_u$); at the limit of proportionality (P$_p$), which represents the end of the region of linear behaviour; and at the ultimate load (P$_u$) (Table 4). Beside the corresponding load, the deflection occurring at midspan ($w_{1,i}$) and the bending stiffness ($k_{1,i}$) are given at each loading step (i = 1/3 for the loading step 0.33 P$_u$, i = p for the loading step P$_p$ and i = u for the loading step P$_u$). The bending stiffness was calculated as $k_{1,i} = P_i/w_{1,i}$. The flexural rigidity was calculated from the displacement $w_1$, and was the highest at C specimens and the lowest at A specimens (Table 4).

**Table 4.** Summary of the experimental results of the specimens of both series of tests (i.e., S1 and S2). The loads ($P_i$) and midspan deflections ($w_{1,i}$) are presented as averaged measured values of all specimens, of the same type and series, and the bending stiffness was calculated as $k_i = P_i/w_i$. In the last six columns comparisons of the different loads ($P_i$), the midspan deflections ($w_{1,i}$), the bending stiffnesses, and flexural rigidity are presented.

| | | At 33% of Ultimate Load | | | At the Proportional Limit | | | At Ultimate Load | | | Comparisons | | | | | | Flexural Rigidity |
|---|---|---|---|---|---|---|---|---|---|---|---|---|---|---|---|---|---|
| | Specimens No. | $P_{1/3}$ (kN) | $w_{1,1/3}$ (mm) | $k_{1,1/3}$ (kN/mm) | $P_p$ (kN) | $w_{1,p}$ (mm) | $k_{1,p}$ (kN/mm) | $P_u$ (kN) | $w_{1,u}$ (mm) | $k_{1,u}$ (kN/mm) | $P_{u,(B,C,D)}/P_{u,A}$ (-) | $P_p/P_u$ (-) | $w_{1,p}/w_{1,u}$ (-) | $k_{1,p}/k_{1,u}$ (-) | $k_{1,1/3}/k_{1,u}$ (-) | $(EI)_{eff}$ (kN*m²) |
| Series S1 | A1–A5 | 6.2 | 19.5 | 0.32 | 12.7 | 39.4 | 0.32 | 18.6 | 67.4 | 0.30 | - | 0.68 | 0.58 | 1.09 | 1.13 | 27.5 |
| | B1–B5 * | 5.7 | 12.1 | 0.48 | 14.3 | 30.9 | 0.46 | 17.1 | 37.5 | 0.46 | 0.92 | 0.84 | 0.82 | 1.01 | 1.04 | 38.7 |
| | C1–C5 | 8.2 | 17.0 | 0.47 | 15.9 | 33.2 | 0.48 | 24.7 | 58.2 | 0.44 | 1.33 | 0.64 | 0.57 | 1.09 | 1.10 | 41.2 |
| Series S2 | A6–A10 | 4.8 | 20.0 | 0.24 | 10.3 | 43.3 | 0.24 | 14.5 | 78.6 | 0.19 | - | 0.71 | 0.55 | 1.26 | 1.29 | 27.3 |
| | B6–B10 | 5.7 | 17.3 | 0.33 | 14.0 | 42.5 | 0.33 | 17.2 | 54.6 | 0.32 | 1.18 | 0.82 | 0.78 | 1.04 | 1.04 | 37.5 |
| | C6–C10 | 7.0 | 20.8 | 0.34 | 13.9 | 40.6 | 0.34 | 21.1 | 80.4 | 0.27 | 1.45 | 0.66 | 0.51 | 1.28 | 1.27 | 39 |
| | D1–D5 | 6.5 | 20.0 | 0.32 | 11.8 | 36.2 | 0.33 | 19.5 | 81.7 | 0.24 | 1.35 | 0.60 | 0.44 | 1.35 | 1.34 | 37.6 |

* B4 was excluded from further analysis due to the change in cross section dimension.

The results show that the ultimate load ($P_u$) and bending stiffness ($k_i$) were the highest in the case of the type C beam specimens, which had the highest percentage of reinforcements, whereas it was the lowest, as expected, in the case of the type A beam specimens, which had no reinforcement (Figure 7). The comparisons of the bending stiffnesses $k_{1,p}$ and $k_{1,u}$ show the plasticity capacity of the tested beams. In the case of the type B beam specimens, both parameters have almost equal values, which corresponds to an almost linear load-deflection relationship up to failure (Figure 7). The biggest difference between these two parameters (i.e., 34%) occurred in the case of the beams D1–D5, which indicates their nonlinear behaviour between the limit of the proportionality and the ultimate load.

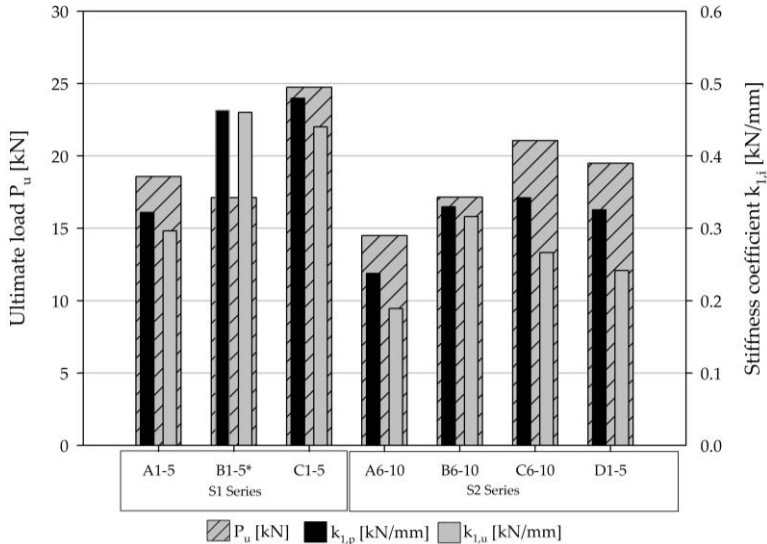

**Figure 7.** Ultimate load-bearing capacity and stiffness coefficient of the tested specimens of types A, B, C, and D.

The average values of the measured midspan deflections were compared at an applied load equal to P = 10 kN, where the load-deflection relationships are still linear (Figure 8).

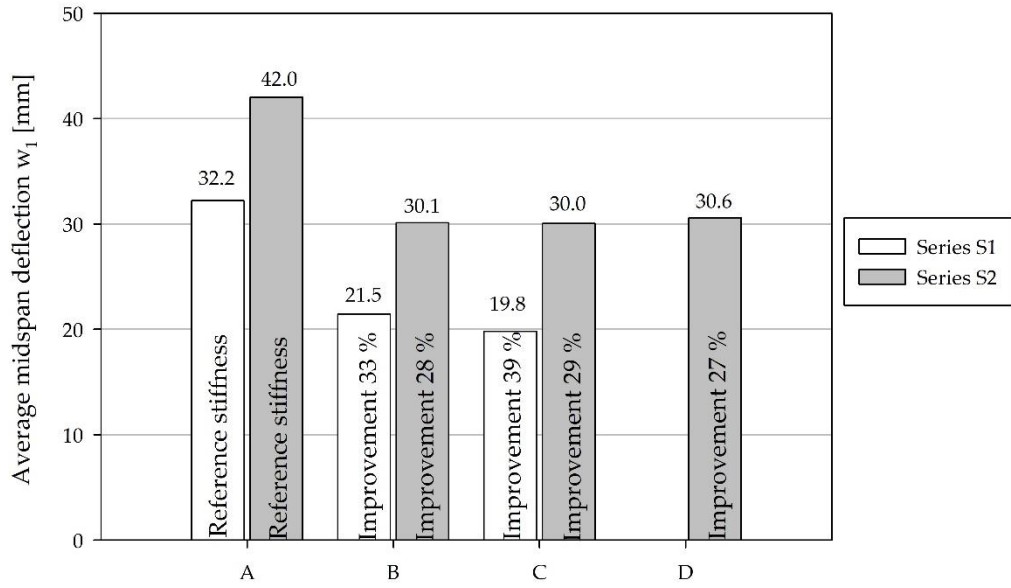

**Figure 8.** Average midspan deflection ($w_1$) of the beam specimens A, B, C, D at the load P = 10 kN and the bending stiffness improvement (%) of the beams of types B, C, D compared to the type A beams.

The average measured midspan deflections ($w_1$—calculated as the mean values of IND 1 and 2) of the beam specimens of types A, B, and C, of Series S1, were 32.2 mm, 21.5 mm, and 19.8 mm, respectively (Figure 8). The bending stiffness improvements of the beam specimens of types B and C when compared to the type A reference specimens, were 33% and 39%, respectively. The average midspan deflections of the Series S2 beams of types A, B, C and D amounted to 42.0 mm, 30.1 mm, 30.0 mm, and 30.6 mm, respectively. The improvement in bending stiffness of the hybrid beams of types B, C, and D, compared to that of type A beams, amounted to 28%, 29%, and 27%, respectively. In both series of tests, the midspan deflection of the reinforced hybrid beams was decreased compared to the reference specimens of type A. The largest standard deviation of the midspan deflection was observed with the type A beams, in the case of both series. This was expected, since these specimens were made solely of wood, which is a non-homogeneous material, with significant variations in density and different grain orientations.

On average, the hybrid beams have improved load-bearing capacity compared to the reference beams (Figure 9). Only the type B hybrid beams, of Series S1, had 8% lower average load-bearing capacity than the reference type A specimens. The reason for this lies in the high shear stresses, which caused a shear failure (Figure 10b), before the flexural load-bearing capacity was reached. The failure happened in the adhesion layer between the wood and the reinforcement. The type C specimens had a load-bearing capacity which was 33% greater than that of the type A reference specimens (Figure 9). The load-bearing capacity of the Series S2 tested specimens of types A, B, C, and D amounted to 14.5 kN, 17.2 kN, 21.1 kN, and 19.5 kN, respectively (Figure 9). The load-bearing capacity was increased in all cases of the reinforced beams by more than 18%. The highest load-bearing capacity was observed in the case of the type C hybrid beams. It was higher than the load-bearing capacity of the type A, B, and D beams by 45%, 23%, and 8%, respectively. Type A, C, and D specimens had a typical tensile failure and type B specimens a typical shear failure (Figure 10).

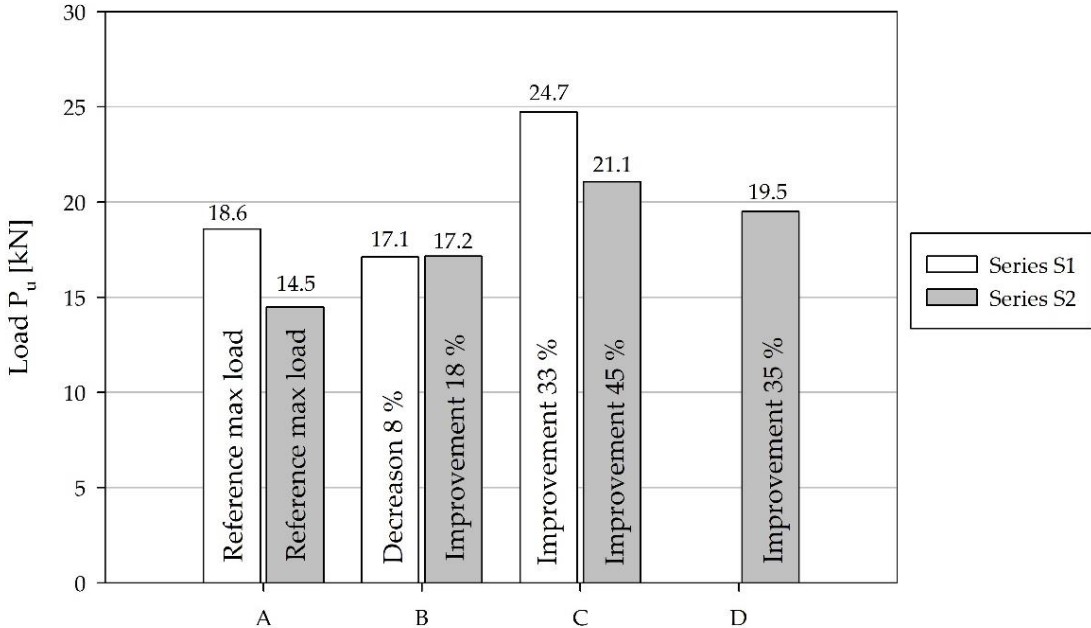

**Figure 9.** Average load-bearing capacity ($P_u$) of the beams of type A, B, C, and D, and the improvement in this capacity of the beams of types B, C, and D compared to the type A beams.

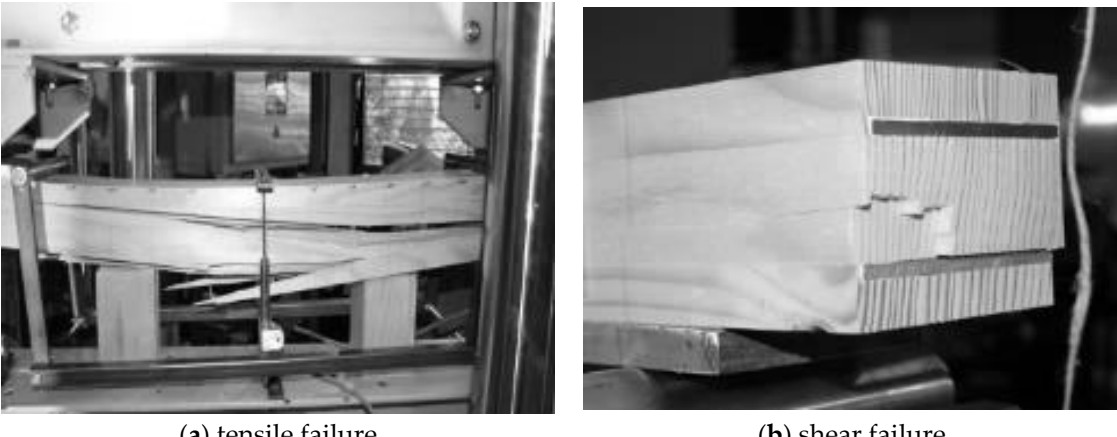

(**a**) tensile failure          (**b**) shear failure

**Figure 10.** Typical failure mechanisms of specimen types A, C, D (**a**) and specimen type B (**b**).

Effective flexural rigidity $(EI)_{eff}$ was calculated on the basis of the average absolute midspan deflection $w_1$, using a well-known expression of technical mechanics (Equation (1)) that corresponds to the four-point bending test load arrangement. The equation neglects a small part of deflections caused by shear stresses.

The hybrid beams of type C had the highest flexural rigidity, followed by the beams of type B, the beams of type D, and lastly the beams of type A (Figure 11). For Series S1 the flexural rigidity of type B and C beams was higher in comparison with beams type A by 41% and 50%, respectively. For Series S2, the flexural rigidity of type B, C, and D beams was higher in comparison with beam type A for 37%, 43%, and 38%, respectively.

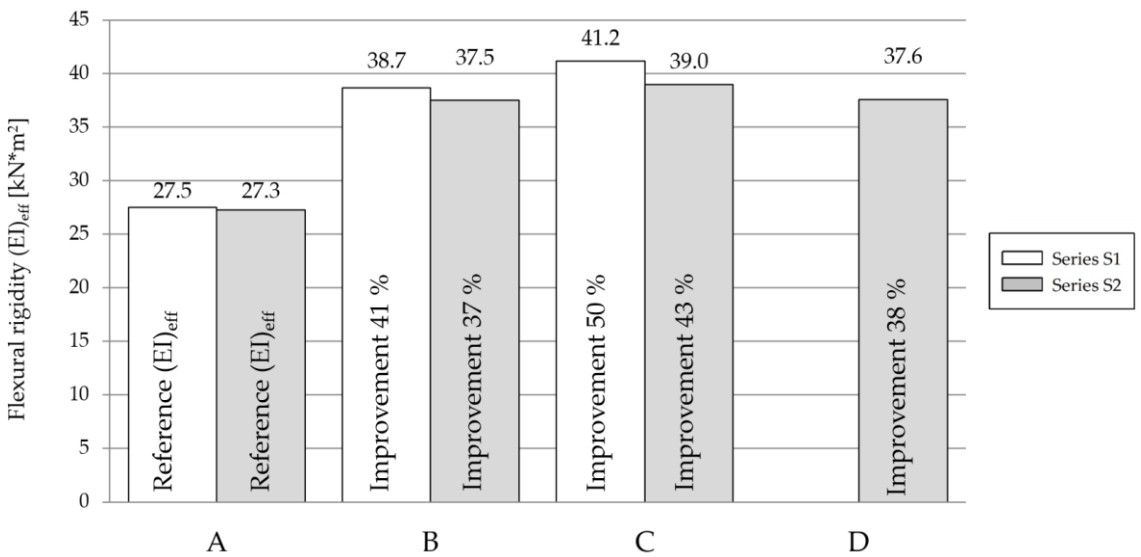

**Figure 11.** Effective flexural rigidity of type A, B, C, and D specimens.

Due to the high ratio between the modulus of elasticity $(E_0)$ and the shear modulus (G) of wood (i.e., $E_0/G = 16$, Table 1) and a relatively high span and specimen height (L/h) ratio (which amounts to L/h = 25.7 and L/h = 27.2 for Series S1 and Series S2, respectively), the deflection caused by shear stresses should not be neglected. Kretschmann [30] suggested that the flexural rigidity should be increased by 10% if the deflection due to the shear stresses is neglected, as it is in the case of Equation (1). The results of a study reported by Eierle and Bös [31] confirmed, that the shear deflection depends on the length and height ratio (i.e., L/h) of the beam, and on the ratio between the shear and elastic moduli (i.e., $G/E_0$). The smaller this ratio is, the larger is the effect of shear stresses on the deflection.

At rectangular shaped simply supported beams with L/h ratio above 25, the shear deflection should not represent more than 2% of the total deflection.

Midspan deflection ($w_1$) was measured in dependence of the load (P), for all beam types of Series S1 and Series S2 (Figures 12 and 13). The angle of the linear part of the curves from the abscissa is defining the bending stiffness; the higher the angle the higher the bending stiffness. Load-bearing capacity is defined by the load at which the curve ends from the exception of specimens marked with * sign, which are specimens, where the full range of at least one LVDT was reached, before the failure occurred.

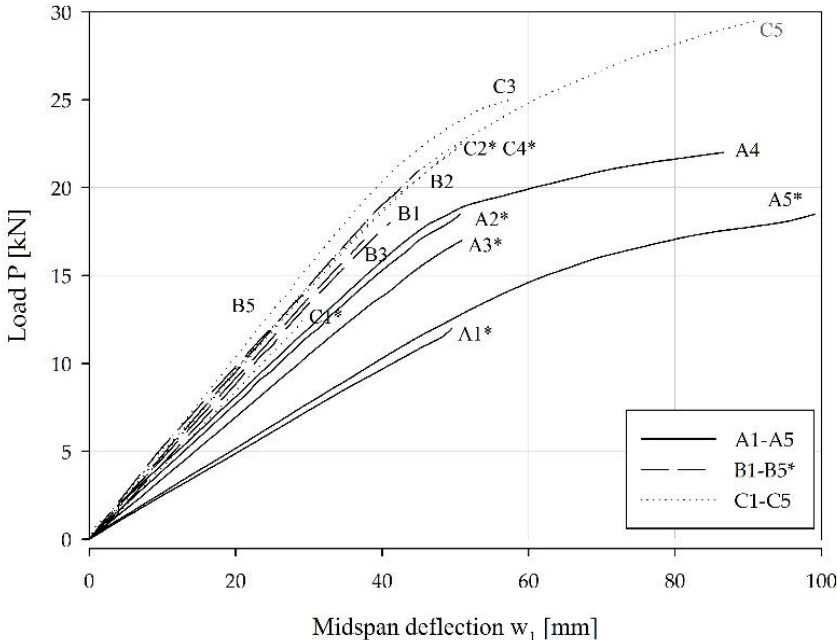

**Figure 12.** Load-midspan deflection relationship of Series S1. * B4 was excluded from further analysis due to the change in cross section dimension.

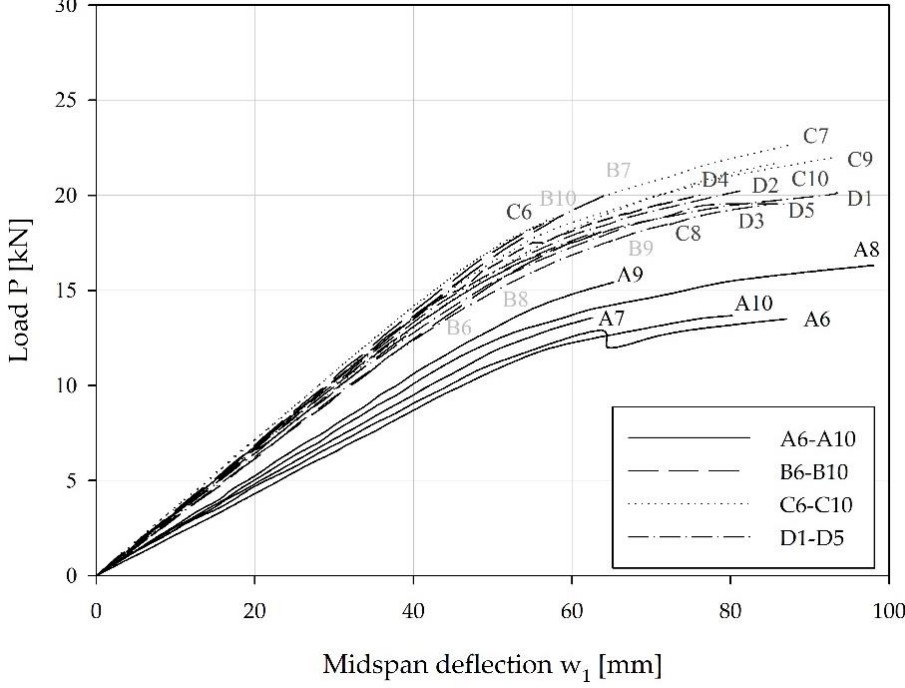

**Figure 13.** Load-midspan deflection relationship of Series S2.

Type A specimens had the smallest bending stiffness and load-bearing capacity but showing the highest dispersion of the results at the same time (Figure 12). The highest values (bending stiffness and ultimate load) were measured at type C specimens. The ultimate load was recorded and taken into consideration in the analysis of the results in Table 4.

In both figures (Figures 12 and 13) the same deflection range and load range had been used in order to be able to directly compare also the behaviour of specimens being tested on two different testing fields. With the increase of the span L and the increase of the distance between the support and the load a, the bending moment has increased (at the same load level) at specimens of Series S2, compared to specimens of Series S1. The increase of the bending moment by 25% in Series S2, has decreased the ultimate load $P_u$ (Series S2) for only 12% compared to ultimate load of specimens of Series S1. The reason lies in a prevailing failure mechanism, which was with Series S2 specimens a typical tensile failure.

## 4. Discussion

Aluminium reinforcements used in wooden beams have improved bending stiffness, load-bearing capacity and flexural rigidity. The experiments showed almost linear dependency between reinforcement-wood ratio and bending stiffness and nonlinear dependency between reinforcement-wood ratio and load-bearing capacity. This was proven also by Kim and Harries [16], who showed that there is a limit of the reinforcement/wood ratio, beyond which the load-bearing capacity can no longer increase. The reason for the smallest increase of load-bearing capacity with specimens B, lies in the high number of shear failure mechanisms (89% of all B specimens tested have failed in shear). Borgin et al. [32] have described the positive effect of the knots in the wood on the shear stress resistance. In the window industry all wood must be free of knots, so the shear stress depends on the wood shear properties only. The shear strength of the specimens could be additionally increased with different wood species in the middle part of the cross section [33], but the most effective it is with the additional reinforcements. There are not many attempts to use aluminium as the reinforcing material. Due to higher moduli of elasticity and better thermal properties, steel is used much more often, [5,9,12]. Nevertheless, in our experimental analysis, it was proven that aluminium is a good material to be used in a first phase of the wood hybrid beam analysis, to define the positioning, orientation of the reinforcements, and the selection of the proper adhesives. One of the biggest advantages of this approach is the ability to process aluminium-wooden hybrid beams with standard machinery for window manufacturing. During the shear strength tests of the adhesives it was observed that surface treatment of the aluminum is essential in order to achieve the adequate adhesion between adhesive and aluminium. The importance of surface treatment of the reinforcement was also declared by Jasienko and Nowak [5], who sanded the steel surface before adhesion. For proper adhesion, aluminium embedded in our specimens was anodized.

The presented study was made as a starting point for further analysis, more focused on the reinforcements in window profiled elements. Mullion profile (middle window profile in the double sashed window, Figure 14) is a combination of one of the most frequently used window profiles and one of the most exposed window elements to wind load.

From the four-point bending tests results and observed failure mechanisms it can be concluded, that the optimal reinforcement arrangement would be a combination of the reinforcement used in the type C and D specimens. With reinforcements positioned in both, the tensile and compressive zones, the tensile and compressive stress strength of the beams (as in the case of the type C specimens) can be increased. A vertically positioned reinforcement over the almost whole cross sectional height (as in the case of the type D specimens) prevents a shear failure before the tensile and compressive stress limits are reached. A preposition of such a window hybrid beam is presented in Figure 15.

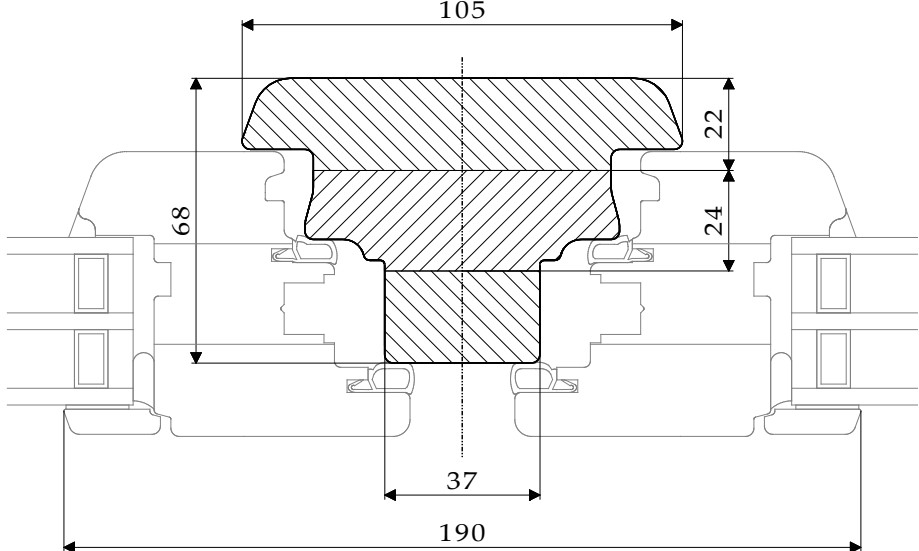

**Figure 14.** Mullion-middle fixed element of the double sashed window (measurements in mm).

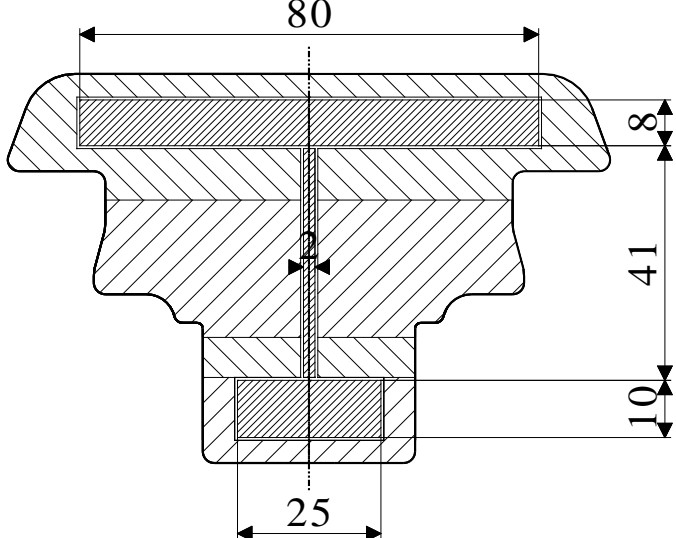

**Figure 15.** Example of window profile with horizontally and vertically positioned reinforcements (measurements in mm).

But on the other hand, a vertical metal reinforcement through the whole specimen height can lead to thermal transmission and therefore decrease thermal efficiency of the window or even worse, can lead to a local decrease of inner temperature that can further on lead to condensation and mold growth [34]. If we could achieve the proper bending stiffness of the window elements with aluminium reinforcements that would not go through the whole height of the profile, this would be the most cost-effective and applicable solution in window production.

In further analysis, the mullion profile will be investigated to define maximal possible bending stiffness and load-bearing improvements, considering positioning, orientation, material, and number of reinforcements.

## 5. Conclusions

An experimental analysis of the effect of reinforcements on the bending stiffness, load-bearing capacity, and flexural rigidity of small cross section hybrid window beams, reinforced with aluminium reinforcements was analysed. The results show that all three analysed parameters were the best in the

case of the type C specimens with 12 (3 × 20 mm) vertical reinforcements in two rows. The type B specimens had a very low load-bearing capacity, due to their low resistance to shear stress, whereas the type D beams never failed due to a lack of shear strength. The shape of the window profile and the results of this analysis indicate that the reinforcement should be arranged by combining the reinforcing methods used in the type C and type D beam specimens. Based on the results of this study, further numerical and experimental analyses will need to be performed, using real shaped window profiles (Figure 15).

**Author Contributions:** B.Š., J.L. and G.F. conceived and designed experiments; B.Š. and J.L. performed the experiments; B.Š. analysed the data; J.L. and G.F. conducted the validation and formal analysis; B.Š., writing—original draft preparation; J.L. and G.F., writing—review and editing.

**Funding:** This work was supported by Ministry of Education, Science and Sport of the Republic of Slovenia, and by the European Regional Development Fund, European Commission (project TIGR4smart, grant number 5441-1/2016/116) and by Ministry of Education, Science and Sport of the Republic of Slovenia within the framework of the Program P2-0182.

**Conflicts of Interest:** The authors declare no conflict of interest.

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
