# Peer review of "Bending Stiffness, Load-Bearing Capacity and Flexural Rigidity of Slender Hybrid Wood-Based Beams"

_forests, doi:10.3390/f9110703_

Round 1
Reviewer 1 Report
Dear Authors,
The manuscript:
‘Bending stiffness of slender hybrid wood-based beams’
is an interesting scientific work presenting new options for wooden beams improvement. The research was very well designed, material for tests was prepared in a right way and finally good tests were provided.
I have two mayor doubts about manuscript text quality: the way of results description and title itself.
Please consider to make direct and specific description of results, writing which tested option had better characteristics rather than writing that ‘characteristics of all specimens are given in table …’. Style wise it doesn’t contribute to the text and it may be considered by readers as not very polite. Basically, it is recommended to write that e.g. ‘specimen C had the best parameters in bending stiffness (Table N)’ rather than writing that ‘parameters of bending stiffness are in Table N’.
Please also note, that title only consists of bending stiffness and in fact more properties were measured, which I think should/could be reflected in the title.
There are several detailed comments in the min text of the manuscript for further consideration when improving whole text.
With best regards,
Reviewer

Author Response
Response to Reviewer 1 Comments:
Please consider to make direct and specific description of results, writing which tested option had better characteristics rather than writing that ‘characteristics of all specimens are given in table …’. Style wise it doesn’t contribute to the text and it may be considered by readers as not very polite. Basically, it is recommended to write that e.g. ‘specimen C had the best parameters in bending stiffness (Table N)’ rather than writing that ‘parameters of bending stiffness are in Table N’.
Response 1:
2. Please also note, that title only consists of bending stiffness and in fact more properties were
measured, which I think should/could be reflected in the title.
Response 2: The title was changed to: Bending siffness, load-bearing capacity and flexural rigidity of slender hybrid wood-based beams.
3. There are several detailed comments in the min text of the manuscript for further consideration when improving whole text.
Response 3: The detailed comments were addressed and all considered in the final version of the new submitted article. The responses to each comment are addresed in the attached pdf. file.

Reviewer 2 Report
The authors focus on an interesting problematic, which tries to analyse the effect of the reinforcement on the flexural stiffness and the bearing capacity of the "hybrid" beams. This problematic is definitely actual nowadays, but from the point of view of journal „Forests“ focus as well as its readers I would expect, if so the composite material and reinforcements are also on the natural basis, i.e. bamboo, cannabis, etc. If, of course, the editor of the magazine does not have a problem with the focus of the article, so everything is all right then. Nevertheless, I would recommend that if a suitable special number is not opened in the „Forests“, move the article to another journal under MDPI, preferably „Materials“.
I would like to make few corrections and additions.
I would move the paragraph on lines 83 to 88 to the chapter Introduction.
The formula on line 258 does not belong to the chapter Results. Move it to the methodology with an adequate description of quantities and units.
Correct the wrong unit (m for mm) in Table 4.
Format the literature according to the requirements of the Forests manual.
I would add:
- the variability (standard deviations or variation coefficients) at least the resulting stiffness, even though the variability is relatively visible from the deflections in Figures 12 and 13,
- with the above-mentioned related statistical evidence of 5 beams within the series (through the quantile of the Student's distribution),
- the legend to Figure 8 is not enough, be careful throughout the text to explain the abbreviations used, eventually consider the inclusion of Abbreviations.
What is really a bit pity that the reinforcements on ligno-cellulosic base has not been studied, just because of a more suitable coefficient of thermal conductivity (vs. aluminium).
Both the results and their explanations are logical, even if in some cases predictable. However, the article is beneficial for deepening of knowledge about the problematics of flexural (bending) rigidity and load-bearing capacity of hybrid beams under specific conditions, especially from the point of view of shear stresses and overall practical use. From my point of view the information about stiffness and the deflections achieved in the elastic area at the level of 1/3 load from the ultimate load, i.e. the real level of the maximum practical load, are important.
Author Response
Response to Reviewer 2 Comments:
1. I would move the paragraph on lines 83 to 88 to the chapter Introduction.
Response 1: The paragraph was moved to Introduction.
2. The formula on line 258 does not belong to the chapter Results. Move it to the methodology with an adequate description of quantities and units.
Response 2: The formula was moved and described.
3. Correct the wrong unit (m for mm) in Table 4.
Response 3: Corrected.
4. Format the literature according to the requirements of the Forests manual.
Response 4: The literature was formated according to the Forests manual.
5. the variability (standard deviations or variation coefficients) at least the resulting stiffness, even though the variability is relatively visible from the deflections in Figures 12 and 13, with the above-mentioned related statistical evidence of 5 beams within the series (through the quantile of the Student's distribution),
Response 5: I have considered to add the standard deviations and other statistical parametrs even before, but there is lack of space in the tabel and would demand even smaller letters, which I think would enable the reader to read the table easily.
6. The legend to Figure 8 is not enough, be careful throughout the text to explain the abbreviations used, eventually consider the inclusion of Abbreviations.
Response 6: I have changed the Legend on Figure 8,9 and 11, to be aligned with the descriptions used in the other parts of the text.
7. What is really a bit pity that the reinforcements on ligno-cellulosic base has not been studied, just because of a more suitable coefficient of thermal conductivity (vs. aluminium).
Response 7: We have considered several choices of using ligno-cellulosic reinforcemens; different (harder) wood species, composites, ... but regarding the small corss-sections we have with window profiles and which need to be reinforced, the modul of elasticity needs to be much higher than the modulus of elasticity of the wood itself (12GPa). Even with the use of aluminum (5,8 times higer elastic modulus), we have achieved maximum 39% of the stiffness improvement, using high percentage of reinforcements (12,9%). Elastic modulus of the hardwood can reach max 20GPa, and of the composites between 20-35GPa.